# Thermal Requirements, Phenology, and Maturation of Juice Grape Cultivars Subjected to Different Pruning Types

**Camilo André Pereira Contreras Sánchez ***, **Daniel Callili**, **Débora Cavalcante dos Santos Carneiro,**
**Safira Pataro Sampaio da Silva**, **Ana Carolina Barduzzi Scudeletti**, **Sarita Leonel** and **Marco Antonio Tecchio**

Department of Horticulture, Campus Botucatu, Faculty of Agricultural Sciences, São Paulo State
University (UNESP), São Paulo 01049-010, Brazil; daniel_callili@hotmail.com (D.C.);
debora.cavalcante-santos@unesp.br (D.C.d.S.C.); safira.pataro@unesp.br (S.P.S.d.S.);
ana.scudeletti@unesp.br (A.C.B.S.); sarita.leonel@unesp.br (S.L.); marco.a.tecchio@unesp.br (M.A.T.)
* Correspondence: camilo.apc.sanchez@unesp.br

**Abstract:** The purpose of this study was to assess the impacts of pruning methods (short and mixed pruning) on the duration of phenological phases and thermal requirements of juice grape cultivars ('Bordô', 'BRS Cora', and 'BRS Violeta'), as well as to confirm the development of chemistry throughout berry ripening. The duration of the following phenological phases was measured in days after pruning over two production cycles: budburst, full-bloom, setting, veraison, and ripening. Degree days were used to compute the thermal requirements. Soluble solids, pH, titratable acidity, and maturation index were all measured as the berries ripened. There was no impact on the length of the phenological cycle or the thermal necessities of the vines due to the various types of pruning. In terms of cultivars, 'BRS Violeta' was found to be earlier than 'Bordô' and 'BRS Cora'. In terms of chemical evolution, the berries achieved 16 °Brix at 21 and 28 days following the veraison, and the greatest values obtained were 17.0 ('BRS Violeta') and 18.4 °Brix ('BRS Cora'). Furthermore, 'BRS Cora' produced more acidic berries. However, regardless of cultivar, the grapes were of high quality, making them a viable option for Brazilian subtropical viticulture.

**Keywords:** subtropical viticulture; BRS Violeta; BRS Cora; Bordô; mixed pruning; short pruning



## 1. Introduction

Brazil's consumption of whole grape juice has climbed by 10% in the previous four years, with per capita consumption reaching 1.36 L in 2020 [1]. To meet this expanding need, one potential option may be to expand grape planting for juice production in non-traditional producing regions, such as the State of São Paulo, which is largely located in subtropical climates. Thus, in order to adopt sustainable cultivation measures in viticulture, studies on cultural management and the introduction of more acclimatized cultivars that meet the needs of the market are essential for viticulture in São Paulo.

'Isabel', Concord', and 'Bordô' are three of the most important juice grape cultivars [2]. However, these traditional cultivars do not adapt well to the subtropical climate of Brazil, resulting in low output and quality [3]. 'Bordô', a rustic cultivar, offers strong resistance to the crop's major fungal diseases, as well as a high concentration of soluble solids and a low acidity [4].

Other cultivars for juice production created in the nation include Embrapa's hybrids 'BRS Cora' and 'BRS Violeta' [5,6]. These cultivars are currently rather popular due to their agronomic and organoleptic properties. 'BRS Cora' has a nice flavor, a high soluble solid content, and is recommended for the enhancement of juices with poor color. It also has high production potential and a medium cycle. 'BRS Violeta', on the other hand, has an early cycle, great production, high sugar content, and low acidity [3]. However, it is crucial to note that numerous factors, such as the types of pruning employed, can impact the features of the vines, and, as a result, the quality of the grape [7,8].

Short, long, and mixed pruning are the three types of pruning employed in viticulture. Short pruning leaves 1 to 3 buds on the producing branches. Long pruning entails leaving 4 to 12 buds on the pruned branches. The combination of short and long pruning on the same plant, altering short and long branches on the same spur, is known as mixed pruning [9]. However, it is stressed that the effect of pruning management may vary depending on cultivar bud fertility and training system [8], necessitating investigations of cultivars in various edaphoclimatic conditions.

Studying the phenology of grapevines is crucial for introducing new cultivars in non-traditional areas as it helps to identify the length of plant growth phases based on climate and seasonal variations [10,11]. Furthermore, knowing the duration of the phenological phases helps to increase the sustainable management practices of the vineyard, as well as in scheduling grape processing firms to receive and process raw material in order to minimize product quality loss [12].

Another useful technique for producers involves the monitoring of the evolution of berry maturity, which allows them to determine the optimum period for harvesting, i.e., when the berries achieve desirable levels of soluble solids and acidity, using chemical components. According to Lima and Choudhury [13], adequate maturity of the vine fruits is vital for the quality of the juices because physical, physiological, and biochemical changes occur during the ripening of the grape.

As a result, the current study sought to assess the phenological behavior, temperature requirements, and maturity progression of the 'Bordô', 'BRS Cora', and 'BRS Violeta' cultivars subjected to short and mixed pruning under subtropical conditions.

## 2. Materials and Methods

### 2.1. Treatments, Experimental Design and Experimental Area

The experimental design was a randomized block design in a $3 \times 2$ factorial scheme (6 treatments), consisting of three juice grape cultivars (Bordô, BRS Cora, and BRS Violeta) and two types of pruning (short pruning and mixed pruning). The experiment was conducted with 96 vines, divided into four blocks, with four plants per plot.

During the summer seasons of 2017 and 2018, the trial took place in an experimental vineyard at the Experimental Farm of the School of Agriculture (FCA) of UNESP, in São Manuel, São Paulo, Brazil (22°44′50″ S, 48°34′00″ W; altitude 765 m). The climate in this region is classified as Cfa by the Köppen classification, indicating a subtropical climate with hot summers. According to the Brazilian soil classification system [14], the soil is a Dystroferric Red Latosol with a sandy texture, with base saturation characteristics greater than 50% in most of the first 100 cm of the B horizon, and is also classified as Hapludox Typical dystrophic. According to the Soil Taxonomy of the USDA (United States Department of Agriculture) [15], the Typical Dystrophic Hapludox presents base saturation characteristics close to 35% up to 150 cm from the B horizon.

Throughout the experiment, a weather sensor 100 m away recorded daily precipitation (mm) as well as maximum, minimum, and average temperatures (°C). During the months of production in the experiment, the lowest average temperature was 17.3 °C in 2017 and 17.6 °C in 2018, while the highest average temperature was 29.6 °C in 2017 and 29.9 °C in 2018. In 2017, the cumulative rainfall was 589 mm, while in 2018, it was 738 mm, with a propensity to concentrate throughout the spring and summer months (Figure 1).

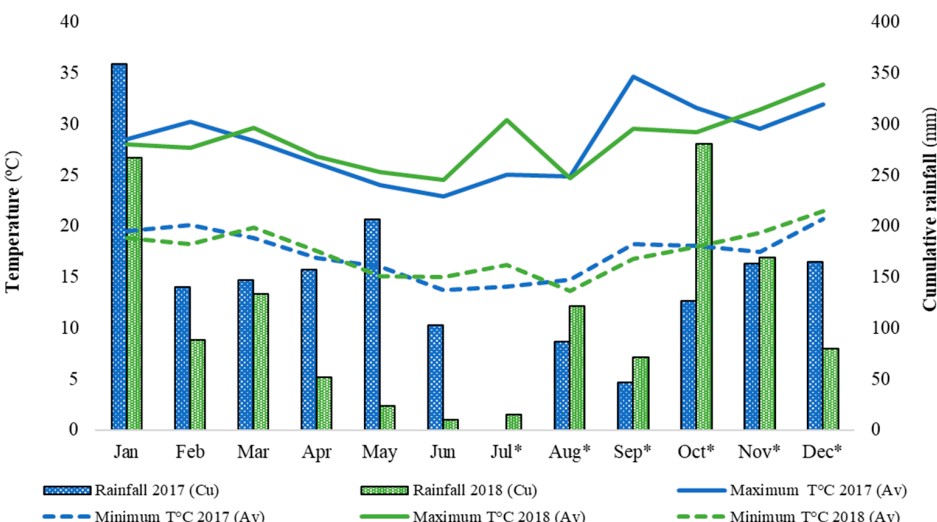

**Figure 1.** The temperature and cumulative rainfall data from the experimental site in São Manuel, located in the state of São Paulo, Brazil, during the productive period of 2017 and 2018. The bars indicate the total amount of rainfall, while the lines represent the minimum and maximum temperatures. * Productive period.

### 2.2. Vine Growing Conditions and Production Pruning

The vines were trained in a low espalier system with wires at 1.0, 1.3, and 1.6 m in height. The row spacing was 2.0 m and the plant spacing was 0.8 m (density of 6250 plants per hectare). The cultivars 'Bordô', 'BRS Cora', and 'BRS Violeta' were grafted onto the 'IAC 572 Jales' rootstock. For irrigation, drippers installed 50 cm from the ground were used, with a higher incidence of use during the budding and fruiting period, as described by Conceição et al. (2017) [16]. An 18% polyethylene screen was utilized for bird protection. Furthermore, all cultural, fertilization, and phytosanitary control was carried out in accordance with conventional regional cultivation methods.

The first and second cycles of production pruning were completed on 22 July 2017 and 20 July 2018, respectively. All production pruning maintained one to two buds per spur in short pruning and three to five buds in mixed pruning. Additionally, 2.5% hydrogen cyanamide was provided after pruning to stimulate and standardize bud sprouting.

### 2.3. Phenology, Thermal Requirement, and Ripening of Grape

The phenological phases were assessed using the criteria suggested by Coombe [17]. Visual observations were made three times a week to assess the length of each phenological stage in days after pruning (DAP). Pruning to budburst, full-bloom, setting, veraison, and full maturation (harvest) are the subperiods.

The harvest dates were set when the berries attained at least 16 °Brix or a minimum ripeness index (SS/AT) of 15. In the first and second cycles, samplings were taken up to 28 and 35 days after the berries began to mature, respectively, because, after this period, the berries began to exhibit signs of rot due to the significant rainfall in December.

For the thermal requirement, the total of the degree day (DD) was estimated from pruning to harvest, that is, the entire cycle, using the Winkler [18] equation:

$$DD = \Sigma \ [\text{average temperature} - 10 \ ^\circ C)] \times \text{days after pruning} \qquad (1)$$

The evolution of soluble solids (SS), pH, titratable acidity (TA), and the maturity index (SS/AT) were used to measure berry ripeness. At the veraison, ten bunches from each plot were randomly picked and assessed until the grapes were fully matured. Every 7 days, that is, at 0, 7, 14, 21, 28, 35 days after the berries began to mature, they were gathered and analyzed.

Direct refractometry of grape must was used to estimate SS via the use of a digital refractometer (Reichert®, model r2i300, Depew, NY, USA), and the findings were represented in °Brix. The pH of the grape must was established by directly reading it (Tecnal® model Tec-10 potentiometer, Piracicaba, Brazil). Titration with 0.1 N NaOH to the equivalency point of pH = 8.2 yielded TA, which was stated as a percentage of tartaric acid. The maturity index was computed as the ratio of SS to TA.

### 2.4. Statistical Analyses

Two production cycles were statistically analyzed. To examine the impacts of cultivars and pruning methods, as well as their interactions, all data were subjected to analysis of variance. The Tukey test at 5% probability was used to compare averages for phenology and thermal requirement, and regression analysis was used to examine the chemical evolution of cultivars during ripening using the statistical program SISVAR®, version 5.7 (Lavras, MG, Brazil).

## 3. Results and Discussion

### 3.1. Phenological Stages and Thermal Requirement

There was no significant interaction ($p > 0.05$) observed between the types of pruning and cultivars (Table 1). The absence of a significant interaction ($p > 0.05$) between the types of pruning and cultivars highlights the need for a thorough examination of the underlying factors influencing the observed results. However, a significant difference was found among cultivars regarding the duration of phenological phases and thermal requirement (Table 1).

**Table 1.** Interaction between phenological phases and degree days (DD) of juice grape cultivars subjected to different types of pruning in two production periods.

| Cultivar/ Pruning | Season | Budburst | | Full-Bloom | | Setting | |
| --- | --- | --- | --- | --- | --- | --- | --- |
| | | Short | Mixed | Short | Mixed | Short | Mixed |
| Bordo | I | 23 ± 1.5 | 22 ± 1.7 b | 52 ± 3.0 | 51 ± 2.0 b | 58 ± 3.1 | 57 ± 2.4 b |
| BRS Cora | I | 25 ± 1.8 | 27 ± 0.6 a | 54 ± 0.6 | 54 ± 0.9 a | 58 ± 0.6 | 61 ± 2.5 a |
| BRS Violeta | I | 23 ± 1.9 | 24 ± 1.8 b | 53 ± 1.4 | 53 ± 1.4 a | 58 ± 1.4 | 59 ± 0.7 b |
| *p*-value | I | 0.29 | | 0.22 | | 0.35 | |
| Bordo | II | 23 ± 1.5 b | 23 ± 1.5 b | 57 ± 3.7 | 57 ± 3.7 | 61 ± 1.6 b | 61 ± 1.6 b |
| BRS Cora | II | 26 ± 1.0 a | 26 ± 2.0 a | 58 ± 3.1 | 58 ± 3.0 | 67 ± 1.5 a | 67 ± 1.4 a |
| BRS Violeta | II | 23 ± 0.8 b | 23 ± 0.6 b | 57 ± 2.3 | 56 ± 3.2 | 64 ± 2.6 b | 64 ± 2.6 b |
| *p*-value | II | 0.90 | | 0.93 | | 0.99 | |

| Cultivar/ Pruning | Season | Veraison | | Harvest | | Full Demands | |
| --- | --- | --- | --- | --- | --- | --- | --- |
| | | Short | Mixed | Short | Mixed | Short | Mixed |
| Bordo | I | 110 ± 2.1 a | 110 ± 1.8 a | 130 ± 4.4 a | 131 ± 2.3 a | 1574 ± 57.6 a | 1588 ± 28.6 a |
| BRS Cora | I | 109 ± 4.7 a | 109 ± 2.3 a | 133 ± 3.5 a | 130 ± 2.9 a | 1601 ± 45.2 a | 1567 ± 39.1 a |
| BRS Violeta | I | 97 ± 2.4 b | 97 ± 1.3 b | 118 ± 3.1 b | 118 ± 2.0 b | 1405 ± 44.2 b | 1408 ± 29.4 b |
| *p*-value | I | 0.99 | | 0.33 | | 0.34 | |
| Bordo | II | 99 ± 1.4 b | 99 ± 2.3 b | 128 ± 1.7 b | 130 ± 2.9 a | 1470 ± 22.3 ab | 1492 ± 38.3 a |
| BRS Cora | II | 105 ± 2.1 a | 104 ± 2.3 a | 133 ± 2.2 a | 131 ± 2.1 a | 1517 ± 61.1 a | 1519 ± 28.9 a |
| BRS Violeta | II | 99 ± 1.5 b | 99 ± 1.6 b | 125 ± 2.3 b | 125 ± 1.2 b | 1440 ± 31.8 b | 1437 ± 14.8 b |
| *p*-value | II | 0.7 | | 0.20 | | 0.65 | |

Data are expressed as mean ± standard deviation (*n* = 6). Values followed by different letters on the same column indicate significant differences (Tukey test, *p* > 0.05).

Specifically, 'BRS Cora' exhibited prolonged durations of phenological phases and accumulated degree days irrespective of the type of pruning applied. The extended durations of phenological phases and increased degree-day accumulation exhibited by 'BRS Cora' suggest that this cultivar may possess unique physiological characteristics or growth patterns that differ from the other cultivars under study. This observation highlights the significance of analyzing these specific components independently (Table 2).

**Table 2.** Phenological stages and degree days (DD) of juice grape cultivars subjected to different types of pruning in two production seasons.

| Phenological Stages (DAP) | Season | Scion | | | | Pruning | | |
|---|---|---|---|---|---|---|---|---|
| | | Bordô | BRS Cora | BRS Violeta | *p*-Value | Short | Mixed | *p*-Value |
| Budburst | I | 22 ± 1.6 b | 28 ± 4.2 a | 23 ± 1.8 b | <0.01 | 23 ± 1.8 b | 25 ± 4.9 a | <0.01 |
| | II | 23 ± 1.4 c | 26 ± 1.5 a | 23 ± 0.7 b | <0.01 | 24 ± 1.9 | 24 ± 2.0 | 0.62 |
| Full-bloom | I | 51 ± 2.6 b | 54 ± 0.7 a | 53 ± 1.3 ab | <0.01 | 53 ± 1.9 | 53 ± 2.1 | 0.46 |
| | II | 56 ± 3.5 ab | 58 ± 2.9 a | 56 ± 2.7 b | <0.05 | 57 ± 3.0 | 57 ± 3.2 | 0.85 |
| Setting | I | 58 ± 2.8 b | 60 ± 2.3 a | 59 ± 1.1 ab | <0.01 | 59 ± 1.9 | 59 ± 3.0 | 0.52 |
| | II | 61 ± 1.6 c | 67 ± 1.4 a | 64 ± 2.5 b | <0.01 | 64 ± 3.0 | 64 ± 3.0 | 0.99 |
| Veraison | I | 110 ± 1.9 a | 110 ± 5.8 a | 97 ± 1.9 b | <0.01 | 106 ± 7.1 | 106 ± 7.4 | 0.97 |
| | II | 99 ± 1.9 b | 104 ± 2.1 a | 99 ± 1.5 b | <0.01 | 101 ± 3.2 | 101 ± 3.0 | 0.93 |
| Harvest | I | 131 ± 3.4 a | 131 ± 3.4 a | 118 ± 2.5 b | <0.01 | 127 ± 7.4 | 126 ± 6.3 | 0.67 |
| | II | 129 ± 2.4 b | 132 ± 2.2 a | 125 ± 1.7 c | <0.01 | 129 ± 4.0 | 129 ± 3.4 | 0.81 |
| Full demands (DD) | I | 1581 ± 44.0 a | 1584 ± 44.1 a | 1406 ± 35.8 b | <0.01 | 1527 ± 100.2 | 1521 ± 86.7 | 0.67 |
| | II | 1480 ± 32 b | 1518 ± 45 a | 1438 ± 24 c | <0.01 | 1476 ± 51.2 | 1481 ± 44.5 | 0.64 |

Data are expressed as mean ± standard deviation (*n* = 6). Values followed by different letters on the same line indicate significant differences (Tukey test, *p* > 0.05).

Regarding the time between pruning and sprouting, a significant difference (*p* < 0.05) was observed only in the first harvest, where vines subjected to short pruning sprouted two days earlier than those subjected to mixed pruning (23.2 versus 25.5 DAP) (Table 2).

In general, in the first season, the average duration of the phases from pruning to budburst, full-bloom, setting, veraison, and full maturation was 24, 53, 59, 106, and 127 days, respectively. The second season took 24, 57, 64, 100, and 129 days from pruning to budburst, full-bloom, setting, veraison, and harvest.

Although short and mixed pruning had no effect on the duration of the three assessed cultivars' phenological phases, Sozim [19] observed that mixed pruning provided precocity in comparison to long pruning. These findings imply that not all grapevine cultivars respond to any method of pruning since bud fertility, which may be described as the ability to distinguish between vegetative and productive buds, is a factor. However, it is important to note that research conducted to evaluate the impact of pruning methods on grapevine phenology is limited.

In all cycles, there was a substantial difference in all phenological phases tested for the cultivars. In general, it was found that the 'BRS Cora' was later in the early phenological stages. In terms of the duration from pruning to harvest, the duration associated with the cultivar 'BRS Violeta' was much shorter than that of 'Bordô' and 'BRS Cora' in the first production cycle, with 118 and 131 days, respectively. 'BRS Violeta' (125 days) demonstrated 3 days of precocity in comparison to 'Bordô' and 7 days in comparison to 'BRS Cora' in the second season (Table 2).

Mariani [20] noticed that the cycle of 'BRS Violeta' took place over 102 days in research conducted in comparable climatic conditions to the ones in the current study, i.e., Cfa climate, therefore recording an earlier timeframe in comparison to the results obtained in the present study. Camargo [21], on the other hand, state that the cycle from pruning to harvesting of 'BRS Violeta' takes 120 days in tropical climatic conditions. It should be noted that the longer season duration of the cv. 'BRS Violeta' reported in this study (118 and 125 days) was attributable to the lower average temperature attained during the assessed production cycles.

Camargo and Maia [5] state that the production cycle for 'BRS Cora' lasts around 130 to 140 days in tropical climates. In the Cfb climate, which is temperate with mild summers, the average duration is 157 days. As a result, the climatic differences that occur in each year and location have a direct impact on the duration of the grapevine cycle, with high temperatures shortening the cycle and low temperatures prolonging it [22,23]. In general, the total thermal requirement of the vines in the first season was 1523 GD and 1478 GD in the second cycle (Table 2). 'BRS Cora' and 'Bordô' required more degree-day accumulation in the first season than 'BRS Violeta', with 1584, 1581, and 1406 GD, respectively. In the second cycle, 'BRS Cora' had the greatest thermal need (1517 GD), while 'BRS Violeta' had the lowest thermal requirement (1438 GD) (Table 2).

According to Ahmed [24], differences in climatic circumstances alter the accumulation of degree days regarding grapes between places and seasons. It is worth noting that this notion is useful for anticipating plant development in a variety of conditions, as well as being a useful indication for studying vine behavior in each production region [25].

### 3.2. Quality Parameter

During grape development, there was no significant interaction between the manner of pruning and the chemical factors ($p > 0.05$) (Figures 2–4). Consequently, the mean of quality parameter for each cultivar was calculated based on the number of days after veraison (Figure 5).

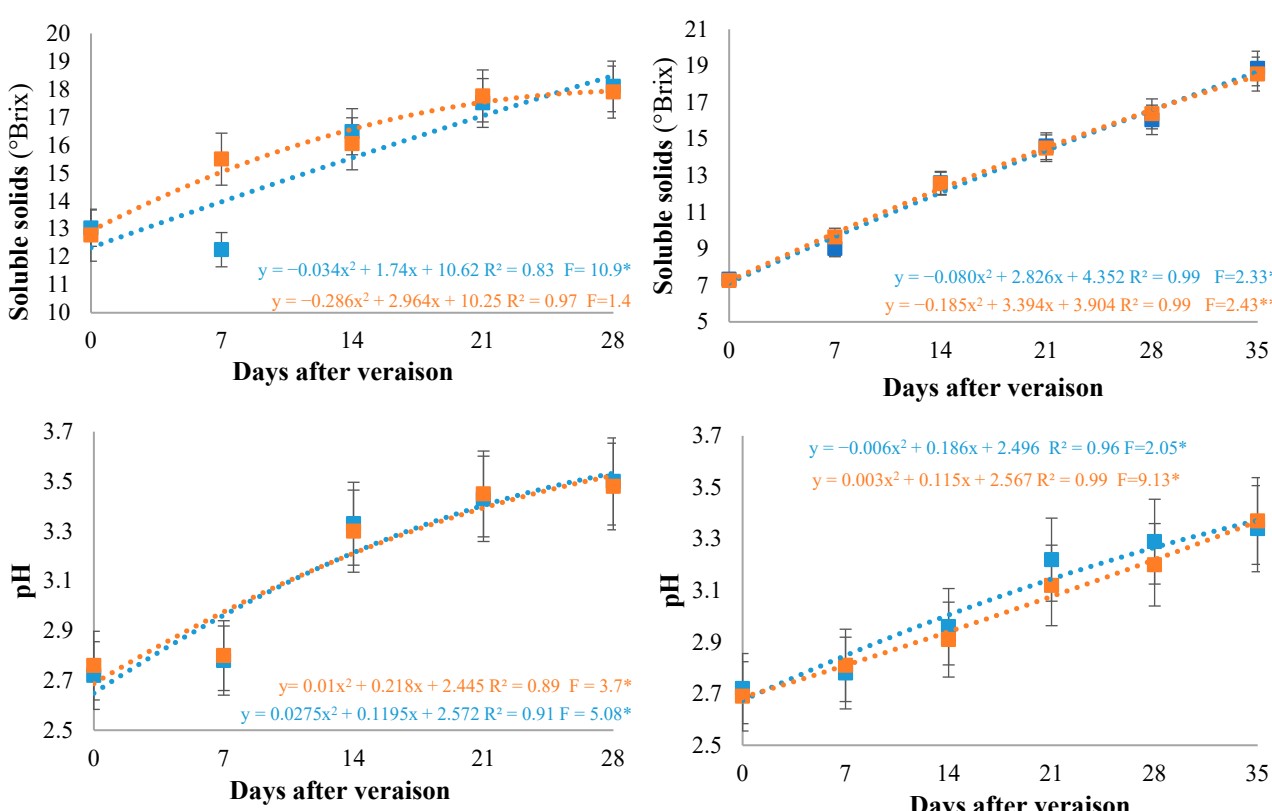

**Figure 2.** *Cont.*

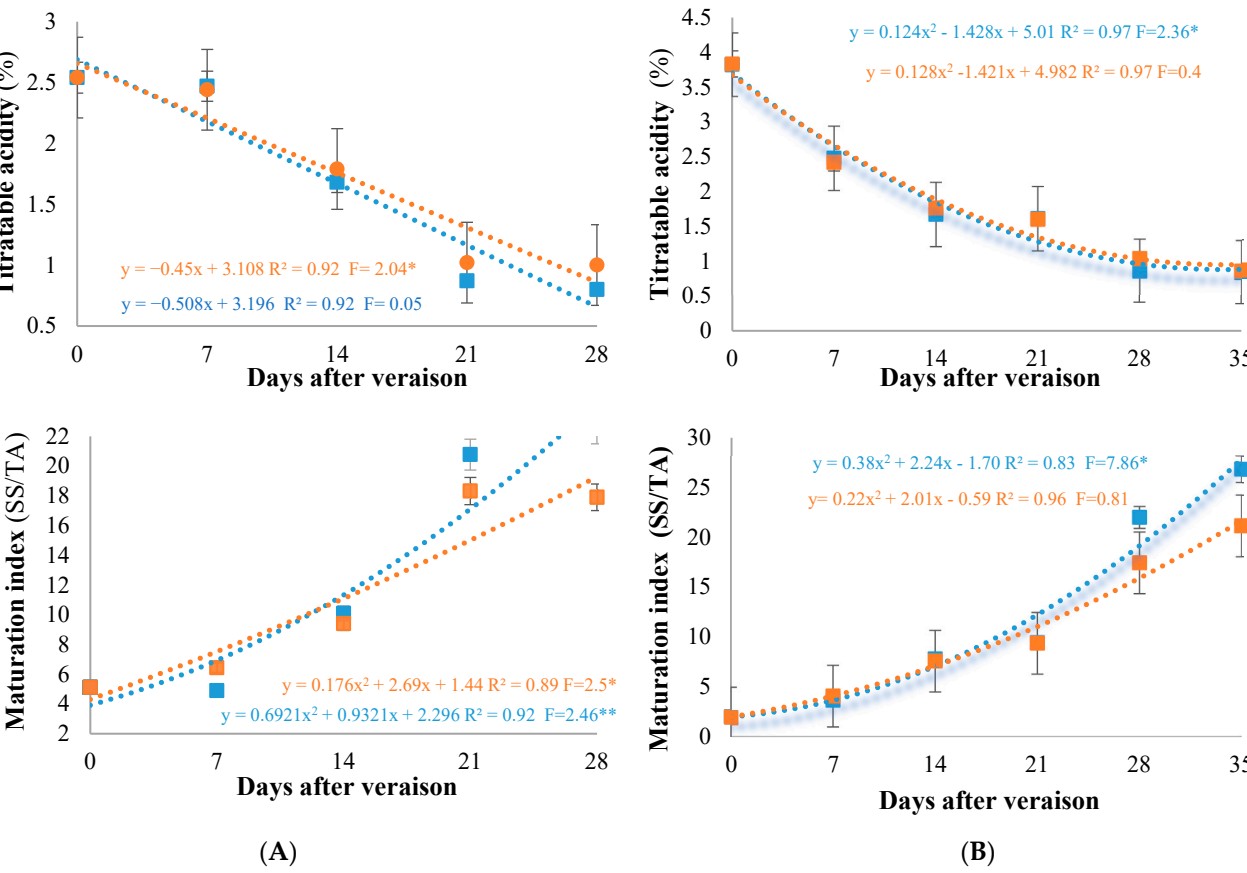

**Figure 2.** The evolution of titratable acidity, pH, soluble solids, and maturation index during ripening of 'Bordo' grapes grown in 2017 (**A**) and 2018 (**B**). Short and mixed pruning are represented by blue and orange colors, respectively. * $p > 0.01$ ** $p > 0.05$. Error bars with 95% CI.

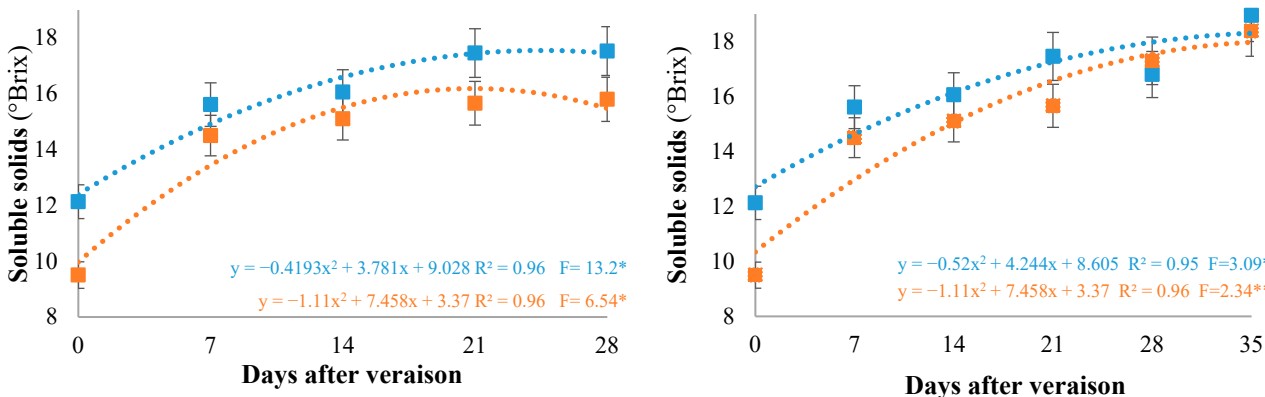

**Figure 3.** *Cont*.

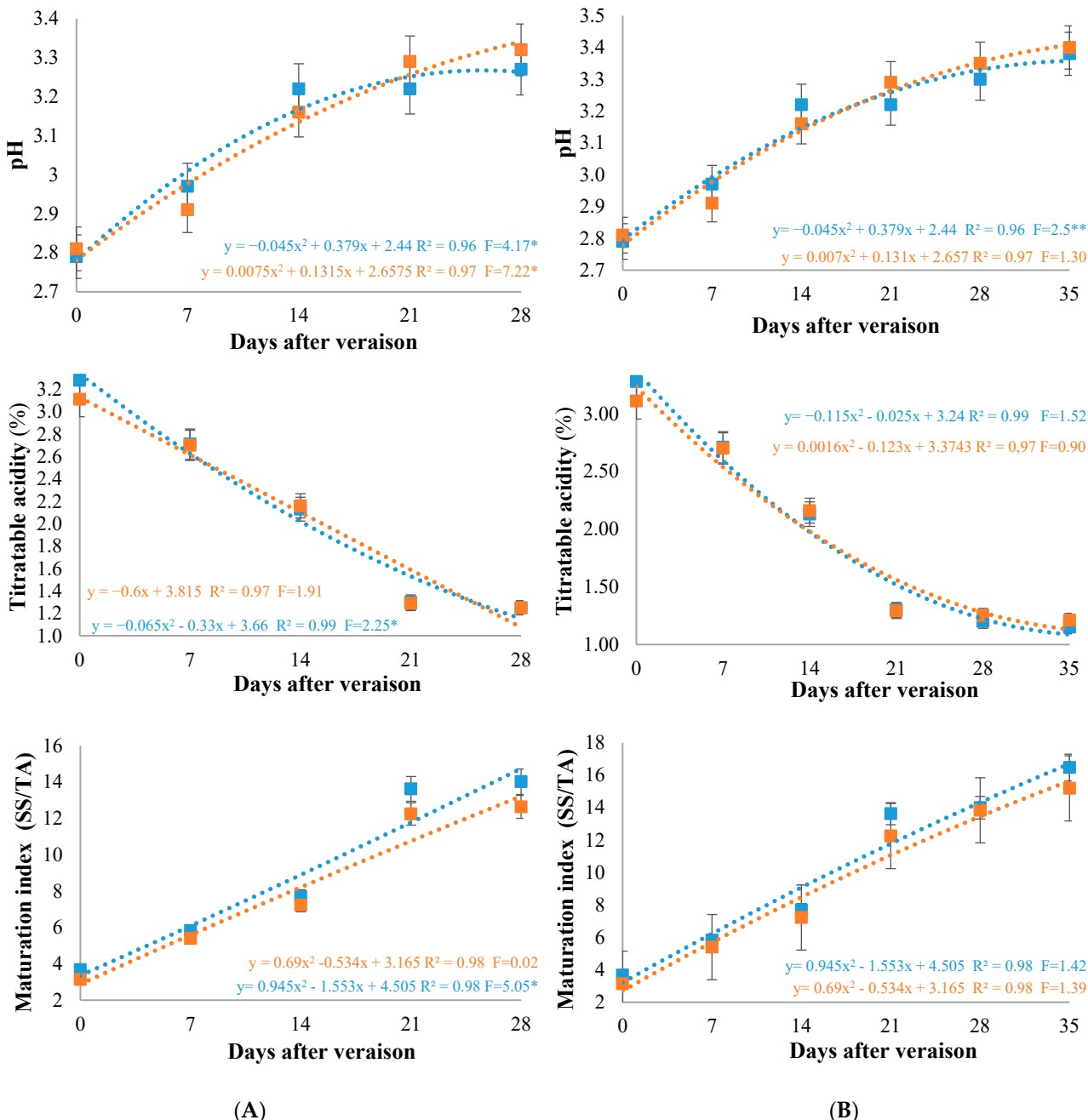

**Figure 3.** The evolution of titratable acidity, pH, soluble solids, and maturation index during ripening of 'BRS Cora' grapes grown in 2017 (**A**) and 2018 (**B**). Short and mixed pruning are represented by blue and orange colors, respectively. * $p > 0.01$ ** $p > 0.05$. Error bars with 95% CI.

According to Brazilian law, the minimum value needed for harvesting grapes for processing is 14 °Brix [26]. However, in the present study, the minimal value determined for collecting was 16 °Brix. In this case, the cultivars 'Bordô', 'BRS Cora', and 'BRS Violeta' reached 16 °Brix 21 days after the veraison, and the maximum point obtained by these cultivars was at 28 days after the veraison, with values of 17.8, 17.9, and 17.4 °Brix, respectively (Figure 5A).

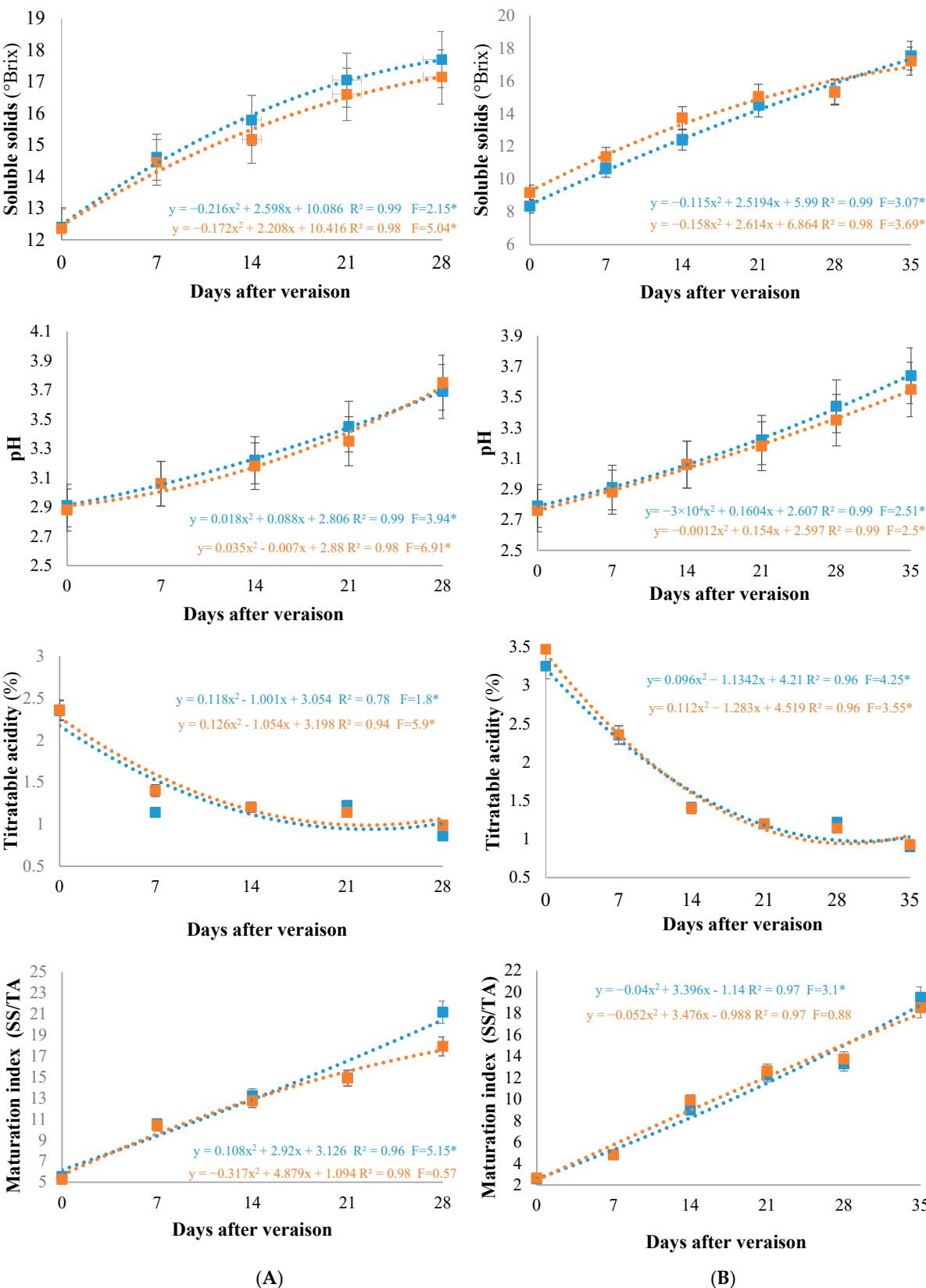

**Figure 4.** The evolution of titratable acidity, pH, soluble solids, and maturation index during ripening of 'BRS Violeta' grapes grown in 2017 (**A**) and 2018 (**B**). Short and mixed pruning are represented by blue and orange colors, respectively. * $p > 0.01$. Error bars with 95% CI.

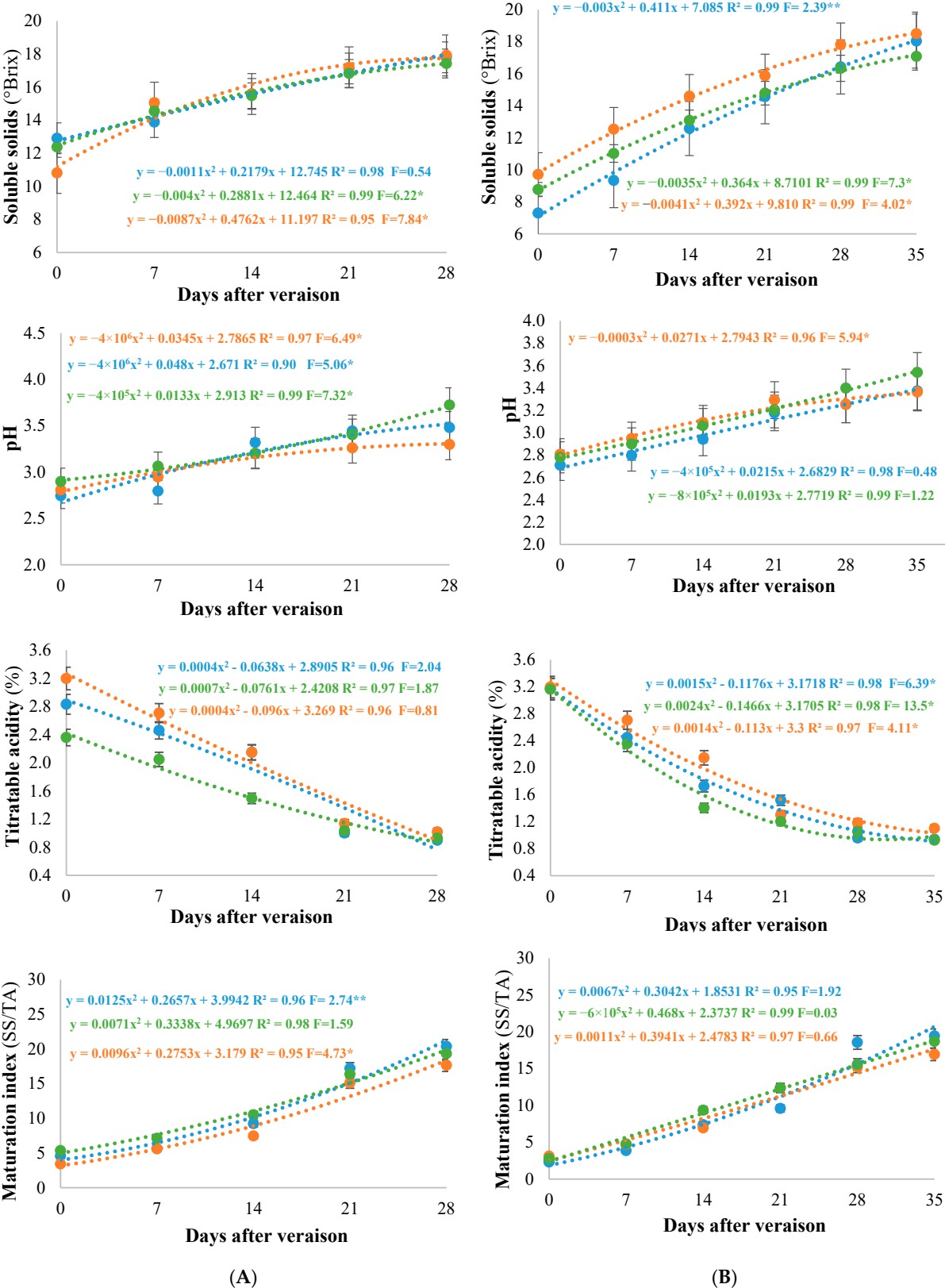

**Figure 5.** The progress of titratable acidity, pH, soluble solids, and maturation index during the ripening of juice grapes cultivated in 2017 (**A**) and 2018 (**B**) 'Bordô', 'BRS Cora', and 'BRS Violeta' are represented by the colors blue, orange, and green, respectively. * *p* > 0.01 ** *p* > 0.05. Error bars with 95% CI.

However, in the second season, 16 °Brix was achieved 28 days after veraison. In this scenario, the highest point of the 'Bordô', BRS Cora', and 'BRS Violeta' cultivars was 18.0, 18.4, and 17.0 °Brix 35 days from the commencement of fruit ripening, respectively (Figure 5B). The soluble solids readings reported for 'BRS Cora' are consistent with the descriptions given by the authors of [5], who state that the soluble solid content of this cultivar should range between 18 and 20 °Brix. However, the 'BRS Violeta' provided lower readings in comparison to the information supplied by the authors of [6], whose descriptions imply that the soluble solid content should be between 19 and 21°Brix.

Ref. [2] reported that 'Bordô' and 'BRS Violeta' had lower amounts of soluble solids than in the current investigation in a study with grape cultivars for processing. When compared to the result achieved in this investigation, the authors of [27] found lower levels for 'BRS Cora'. The percentage of sugar in grapes is inextricably tied to the quality of the grape must used to make juices and wines [2]. As a result of the high number of soluble solids identified in the current study, it is proposed that these cultivars shown good adaptability to the local edaphoclimatic conditions.

In general, the pH of the cultivars ranged from 3.30 (BRS Cora) to 3.72 (BRS Violeta) 28 days after the start of ripening in the first season (Figure 5B). In turn, 35 days after veraison, the pH values found in the second season were 3.36 for 'BRS Cora', 3.37 for 'Bordô', and 3.54 for 'BRS Violeta' (Figure 5B). The rise in pH in the berries was caused by the salinization of organic acids and an increase in potassium cation [28].

Although pH is not required by Brazilian law, it is an important characteristic to examine since it is directly connected to anthocyanin stability and the color intensity of grape juice or red wine [29,30]. Furthermore, the determination of pH in grapes for juice is an essential component since, when the value is low, it helps balance the sweet and acidic flavors [31].

At 28 days after veraison, the titratable acidity ranged from 0.90 to 1.02% in the first season (Figure 5A). In the second season, 35 days following the veraison, the titratable acidity levels varied from 0.93 to 1.10% (Figure 5B). In this scenario, 'BRS Cora' had more acidic berries, that is, higher titratable acidity and lower pH, than 'Bordô' and 'BRS Violeta' in both seasons.

The values observed in this study are higher on average than those discovered by the authors of [2] in 'Bordô' (0.90 versus 0.53%) and 'BRS Violeta' (0.93 versus 0.73%). However, for 'BRS Cora', the observed results are lower compared to those reported by the authors of [12,27], who conducted their studies in a tropical climate. Several physiological processes contribute to the decrease in titratable acidity (TA) during grape ripening. However, temperature variation, light intensity, and rainfall can also impact this reduction, leading to variations in vine metabolism. These factors can either promote or impede the genetic potential of the grapes [32].

Because the maturation index is generated from the connection between sugar and acidity levels (SS/AT), the low acidity of 'Bordô' and 'BRS Violeta' resulted in higher maturation index values, with roughly 20.3 and 19.3 in the first season and 19.4 and 18.7 in the second season, (Figures 5A and 5B) respectively. However, 'BRS Cora' is distinguished by its ability to retain high amounts of soluble solids and acidity until full maturity, as documented by various scholars [12,27,32].

Given that the maturation index identifies the optimal time for harvesting, that is, when there is a better balance of sugars and acids, and that Brazilian legislation requires that the range for grapes destined for juice processing be between 15 and 45, it is recommended that the harvest of these cultivars takes place between 21 and 28 days after the veraison, which may vary depending on the climatic conditions of each harvest.

Considering the fact that grape juice quality is linked to the chemical quality of the berries, the current study confirmed that 'Bordô,' 'BRS Cora', and 'BRS Violeta' had acceptable chemical characteristics, making them a suitable choice for Brazilian subtropical viticulture.

Considering the results of the study and the specific context in which the research was carried out, it is advisable to interpret the results with caution. Although the study provides information on the thermal requirements, phenology, and maturation of the vines, it is essential to recognize the potential influence of external factors, such as climatic conditions and application rates of hydrogen cyanamide, on the observed results.

## 4. Conclusions

The duration of the phenological phases and the thermal requirements of the juice grape cultivars were unaffected by the different pruning types. This is important for sustainable viticulture, as it allows for the adoption of pruning practices that are more appropriate to the local context without compromising the development of culture.

There is a need to expand studies on the different pruning methods in grape cultivars intended for processing purposes, as well as in emerging grape-growing regions, such as those with subtropical climatic conditions, in order to comprehend their influences on phenology, thermal requirements, and maturation.

In comparison to 'BRS Cora' and 'Bordô', 'BRS Violeta' was the most precocious cultivar; this is useful for producers who need to anticipate the harvest and commercialization of grapes, which can be important in terms of competitiveness.

The cultivar with the highest soluble solid content, highest acidity, and lowest pH was BRS Cora'. This is very important for the juice industry, as fruits with these characteristics are needed for the production of high-quality juices.

The soluble solid concentration of the 'Bordô', 'BRS Cora', and 'BRS Violeta' cultivars was over 16 °Brix, indicating acceptable chemical quality.

Harvesting the grapes between 21 and 28 days after veraison is recommended in order to imbue them with a better balance of sugars and acids.

**Author Contributions:** Conceptualization, M.A.T.; methodology, C.A.P.C.S.; software, D.C.; validation, M.A.T. and S.L.; formal analysis, C.A.P.C.S., D.C., D.C.d.S.C., S.P.S.d.S. and A.C.B.S.; investigation, C.A.P.C.S., D.C., D.C.d.S.C., S.P.S.d.S. and A.C.B.S.; resources, M.A.T.; data curation, C.A.P.C.S. and D.C.; writing—original draft preparation, C.A.P.C.S. and D.C.; writing—review and editing, C.A.P.C.S., D.C., S.L. and M.A.T.; visualization, M.A.T. and S.L.; supervision, M.A.T.; project administration, M.A.T.; funding acquisition, M.A.T. All authors have read and agreed to the published version of the manuscript.

**Funding:** This research was funded by the State of São Paulo Research Foundation (FAPESP) (process nos. 2015/16440-5; 2020/12152-3); the Research Productivity Grant (process no. 307377/2021-0); and the Coordination for the Improvement of Higher Education Personnel (CAPES) through scholarship.

**Data Availability Statement:** Not applicable.

**Acknowledgments:** The authors would like to thank the State of São Paulo Research Foundation (FAPESP) for their financial support (process nos. 2015/16440-5; 2020/12152-3); CNPq for the Research Productivity Grant (process no. 307377/2021-0); and the Coordination for the Improvement of Higher Education Personnel (CAPES) for the scholarship. The authors would also like to thank the fruit growing research group (FRUT!) for their assistance, as well as São Paulo State University (UNESP), Faculty of Agricultural Sciences, Campus Botucatu for their help in carrying out the study.

**Conflicts of Interest:** The authors declare no conflict of interest.

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
