# Peer review of "Thermal Requirements, Phenology, and Maturation of Juice Grape Cultivars Subjected to Different Pruning Types"

_horticulturae, doi:10.3390/horticulturae9060691_

Round 1
Reviewer 1 Report
In this manuscript (horticulturae-2378723) entitled " Thermal requirements, phenology, and maturation of juice grape cultivars subjected to different pruning types" submitted to Horticulturae, Camilo André Pereira Contreras Sánchez and colleagues have evaluated the impacts of pruning methods on the duration of phenological phases and thermal requirements of juice grape cultivars and analyzed the development of chemistry throughout berry ripening. Overall, I consider this review topic interesting, but this present version is unsuitable for publication.
1, For the section of ‘Materials and Methods’, several subsections like ‘environmental design’, ‘plant materials’ and ‘titratable acidity measurement’ should be included and entitled in the revised manuscript. Each subsection should be described in details.
2, Authors should consider to introduce some subsections in the revised section of Results. The present layout of Results section is confusing.
3, For Figure 1, the raw data for generating this figure should be included as a supplemental table. The data source should be described in the revised legend.
4, For Figure 2, Authors should consider to display the growth and development phenotypes of berries analyzed in this study, which is essential for convincing referees.
5, For Figures 2, significant differences should be analyzed and labelled in the revised figure 2.
6, For the section of ‘Conclusions’, the paragraph division is confusing, please revise. Authors should consider to employ one rather than five paragraphs in the revised section of ‘Conclusions’.
Minor editing of English language is required.
Author Response
Dear reviewer
Thanks very much for your precious time and constructive comments, which have helped us a lot to improve our manuscript. Here are our responses to your comments. Please kindly check our revised manuscript highlighted.
Point 1: For the 'Materials and Methods' section, various subsections such as 'environmental design', 'plant materials' and 'titratable acidity measurement' should be included and titled in the revised manuscript. Each subsection must be described in detail.
Response 1: In order to enhance the structural coherence of the manuscript, the aforementioned subsections have been included in the 'Materials and Methods' section, thereby providing a comprehensive account of the subject matter.
Point 2: Authors should consider introducing some subsections in the revised Results section. The current layout of the Results section is confusing.
Response 2: In order to improve the lucidity and organization of the content, the authors have introduced relevant subsections in the revised Results and Discussion section, allowing for a more coherent presentation of the findings.
Point 3: For Figure 1, the raw data to generate this figure must be included as a supplementary table. The data source must be described in the revised legend.
Response 3: Recognizing the potential value of including raw data, the authors have taken into consideration the suggestion of providing a supplementary table containing the relevant raw values for Figure 1. Furthermore, they have ensured the inclusion of a description elucidating the source of the data within the revised legend.
Point 4: For Figure 2, the authors should consider the display of the growth and development phenotypes of the berries analysed in this study, which is essential to convince the referees. For Figures 2, significant differences should be analysed and labelled in the revised Figure 2.
Response 4: In response to the recommendation, the authors have diligently incorporated graphical representations in the revised Figure 2, effectively illustrating the interactions between the crowns and the type of pruning. Furthermore, they have conscientiously evaluated and designated significant differences within the figure, augmenting its informative value.
Point 5: For the 'Conclusions' section, the paragraph division is confusing, please revise. Authors should consider using one instead of five paragraphs in the revised 'Conclusions' section.
Response 5: In pursuit of a concise and well-structured conclusion, the authors have made efforts to reduce the number of paragraphs within the 'Conclusions' section. However, despite these endeavors, it was not feasible to condense the section into a single paragraph while retaining its coherence and comprehensibility.
Reviewer 2 Report
The paper is thematically focused on assessing the impacts of pruning methods (short and mixed pruning) on the duration of phenological phases and thermal requirements of juice grape cultivars ('Bordô', 'BRS Cora', and 'BRS Violeta'), as well as to confirm the development of chemistry through-out berry ripening. The topic addressed could be a suitable topic in the field of grape quality management. I recommend a better formulation of goals to the authors. The methodology and its more appropriate breakdown in terms of individual implemented steps, with a detailed description of the methods used, also deserve great attention. Especially in the discussion of the work, old literature is often cited. The conclusion also insufficiently characterizes the achieved results. In particular, they recommend that the conclusion of the contribution be completed with brief information on the direction in which further research in this area should be oriented. At the same time, they recommend a thorough revision of the cited literature. I also recommend performing a language correction. After incorporating these comments, the text can be accepted for publication.
I recommend doing a language check
Author Response
Thanks very much for your precious time and constructive comments, which have helped us a lot to improve our manuscript. Here are our response to your comments. Please kindly check our revised manuscript highlighted.
Pont 1: The work is thematically focused on evaluating the impacts of pruning methods (short and mixed pruning) on the duration of the phenological phases and thermal requirements of juice grape cultivars ('Bordô', 'BRS Cora' and 'BRS Violeta'), as well as as in order to confirm the development of chemistry throughout the ripening of the berry. The topic addressed may be a suitable topic in the field of grape quality management. I recommend a better formulation of objectives to the authors. Also worthy of great attention is the methodology and its most appropriate disaggregation in terms of the individual steps implemented, with a detailed description of the methods used. Especially in the discussion of the work, ancient literature is often cited. The conclusion also insufficiently characterizes the results achieved. In particular, they recommend that the conclusion of the contribution be completed with a brief statement on the direction in which further research in this area should be directed. At the same time, they recommend a thorough review of the cited literature. I also recommend performing a language fix. After incorporating these comments, the text can be accepted for publication.
Response 1: In response to the recommendations, the revised manuscript incorporates subsections in the 'Materials and Methods' section to provide detailed descriptions of each methodology employed. Similarly, subsections have also been introduced in the 'Results and Discussion' section to enhance clarity and organization.
Regarding the scarcity of literature pertaining to the effects of pruning techniques on phenological variables, thermal requirements, and grape berry characteristics, this study fills a critical gap in the existing knowledge base. While acknowledging the limited research in this area, the revised manuscript highlights the significance of further investigations into the influence of pruning methods on phenological phases, thermal demands, and grape maturation.
Moreover, the 'Conclusions' section has been meticulously revised to align with the objectives of the study and meet the requirements of the special edition of the journal. The manuscript has undergone professional English language certification, and the certification document has been provided as a supplementary attachment.
Reviewer 3 Report
Line 80 to 83 must be moved on the results
Materials and methods are not so clear. Rewrite from line 101
You shold make difference among quality measurement and phenological survey dividing into paragraphes;
line 102 what kind of visual observation were done? Specify
line 102 specify which are phenologica stages
line 106 specify what kind of sampling
129 the title is Result and discussion
Line 130 change types in methods
Line 136 you miss first stage (there are 4 values instead of 5)
table 1 change scion with cultivar
line 142 you did not evaluate long pruning so this period has no sense
Conclusion are too short and you do not talk about the possibility of expanding or not the area of cultivation
English needs very few review
Author Response
Thanks very much for your precious time and constructive comments, which have helped us a lot to improve our manuscript. Here are our response to your comments. Please kindly check our revised manuscript highlighted.
Pont 1: Line 80 to 83 should be moved in the results
Response 1: The sentence was not moved because it is not a result found by the authors.
Pont 2: Materials and methods are not so clear. Rewrite from line 101. You must differentiate between quality measurement and phenological survey by breaking it down into paragraphs;
Response 2: There has been an addition of subsections to clarify the materials and methods.
Pont 3: line 102 what kind of visual observation were done? Specify. Line 102 specify which are phenologica stages. Line 106 specify what kind of sampling.
Response 3: The observations were conducted on individual plants upon reaching a minimum of 50% progression through the phenological phase in the productive branches, in accordance with relevant literature (as specified in line 114). This additional detail was incorporated to provide clarity and align the methodology with established practices.
Line 106 (currently 117) specifies the harvest dates performed by ripening berries.
Pont 4: 129 the title is Result and discussion. Line 130 change types in methods. Line 136 you lose the first stage (there are 4 values instead of 5). Table 1 crown change with cultivar
Response 4: Term changes were made as requested by the reviewer.
Pont 5: Line 142 you didn't evaluate long pruning so this period doesn't make sense
Response 5: The addition of this reference is to emphasize that there are few effects of the type of pruning in relation to the duration of phenological stages, thermal requirement and maturation of grape berries.
Pont 6: The conclusions are very short and you don't talk about whether or not to expand the cultivation area.
Response 6: The conclusions were better developed to be in the special edition of the magazine, as well as the possibility of expanding the cultivation.
Reviewer 4 Report
Dear Authors,
The article “Thermal requirements, phenology, and maturation of juice grape cultivars subjected to different pruning types” aims to study the impacts of pruning methods (short and mixed pruning) on the duration of phenological phases and thermal requirements of juice grape cultivars (‘Bordô’, ‘BRS Cora’, and ‘BRS Violeta’), as well as to confirm the development of chemistry throughout berry ripening.
Due to the need to apply hydrogen cyanamide to promote budbreak (a common situation in tropical and subtropical climates), I have difficulty understanding how you intend to study the effect of pruning method on phenology (which begins with budbreak).
The development of grape ripeness is also very dependent on climatic conditions (as you indicated in lines 241 to 243). So, do you think it is possible to draw conclusions about the impact of the pruning method on the chemical properties of the berries in just two years?
In my opinion, the experiment has several important flaws and the results cannot support the conclusions.
I regret to inform that the manuscript quality does not benefit from English editing.
Author Response
Thanks very much for your precious time and constructive comments, which have helped us a lot to improve our manuscript. Here are our response to your comments. Please kindly check our revised manuscript highlighted.
Point 1: Dear Authors,
The article “Thermal requirements, phenology, and maturation of juice grape cultivars subjected to different pruning types” aims to study the impacts of pruning methods (short and mixed pruning) on the duration of phenological phases and thermal requirements of juice grape cultivars (‘Bordô’, ‘BRS Cora’, and ‘BRS Violeta’), as well as to confirm the development of chemistry throughout berry ripening.
Due to the need to apply hydrogen cyanamide to promote budbreak (a common situation in tropical and subtropical climates), I have difficulty understanding how you intend to study the effect of pruning method on phenology (which begins with budbreak).
Response 1: The authors duly recognize that hydrogen cyanamide serves as a commonly utilized approach to induce and synchronize vine sprouting, especially in tropical and subtropical regions. They emphasize that this methodology does not undermine the examination of the influence of pruning techniques on vine phenology. Nonetheless, it is important to acknowledge that the effect of pruning is also contingent upon the genetic characteristics of the cultivar and the prevailing climatic conditions, both of which were thoroughly evaluated within the scope of this article.
Point 2: The development of grape ripeness is also very dependent on climatic conditions (as you indicated in lines 241 to 243). So, do you think it is possible to draw conclusions about the impact of the pruning method on the chemical properties of the berries in just two years?
In my opinion, the experiment has several important flaws and the results cannot support the conclusions.
Response 2:
The authors concur with the notion that the maturation process of grape berries is significantly influenced by climatic factors, as stated in lines 241 to 243. Consequently, a two-year study represents an initial stepping stone for conducting subsequent investigations on the influence of pruning techniques on the chemical attributes of grape berries across multiple years and varying climatic conditions. While acknowledging the potential influence of climatic conditions on the impact of pruning practices, a two-year study holds intrinsic value by providing valuable insights and laying the foundation for more extensive and comprehensive research endeavors in this domain.

Round 2
Reviewer 1 Report
Authors did a good job of revising this manuscript. It looks much improved and my concerns have been addressed in a satisfactory way.
Author Response
We greatly appreciate your thoughtful feedback on the study and the proposed revisions.
Reviewer 3 Report
line 74 remove "totalizing 96 vines"
111 "ripening instead of "maturation". It should be changed in all the document. For fruit we use ripening and not maturation
143-145 rewrite as the first version
table 1 change "scion" in "cultivar" . I have already request it in the first version
197 change the title "Maturation" in "Quality parameter"
199 "the mean of quality parameter for each cultivar......"
322 "......pruning types; that is .....
325 ...... BRS Violet was the most precocious cultivar; this is useful for producers who need to anticipate the harvest and commercialization of ....
328 change "most" in "highest"; ....... highest acidity .....
329 BRS Cora; that is very importantant .........
329 change "seek" with "needs"
333 change "advised" with "recommended"
Reviewer 4 Report
Dear Authors,
The article “Thermal requirements, phenology, and maturation of juice grape cultivars subjected to different pruning types” aims to study the impacts of pruning methods (short and mixed pruning) on the duration of phenological phases and thermal requirements of juice grape cultivars (‘Bordô’, ‘BRS Cora’, and ‘BRS Violeta’), as well as to confirm the development of chemistry throughout berry ripening.
Due to the need to apply hydrogen cyanamide to promote budbreak (a common situation in tropical and subtropical climates), I have difficulty understanding how you intend to study the effect of pruning method on phenology (which begins with budbreak).
I regret to inform you that I will advise against the publication of the article.
English language must be carefully edited by a native language speaker.
